# SonicSense:
# Object Perception from In-Hand Acoustic Vibration

**Jiaxun Liu**  **Boyuan Chen**
Duke University
http://generalroboticslab.com/SonicSense

**Abstract:** We introduce SonicSense, a holistic design of hardware and software to enable rich robot object perception through in-hand acoustic vibration sensing. While previous studies have shown promising results with acoustic sensing for object perception, current solutions are constrained to a handful of objects with simple geometries and homogeneous materials, single-finger sensing, and mixing training and testing on the same objects. SonicSense enables container inventory status differentiation, heterogeneous material prediction, 3D shape reconstruction, and object re-identification from a diverse set of 83 real-world objects. Our system employs a simple but effective heuristic exploration policy to interact with the objects as well as end-to-end learning-based algorithms to fuse vibration signals to infer object properties. Our framework underscores the significance of in-hand acoustic vibration sensing in advancing robot tactile perception.

**Keywords:** Tactile Perception, Object State Estimation, Audio

## 1  Introduction

By shaking a container, we can tell its inventory status from the generated acoustic vibrations, such as the quantity and geometry of the objects inside. Similarly, we can identify the material and geometry of the entire object through multiple tappings. Human hands are equipped with high-frequency skin vibrations to help capture such complex object properties [1]. However, despite the significance of acoustic vibrations for tactile perception, equipping robot manipulators with acoustic vibration sensing capability for rich object perception remains difficult [2, 3, 4, 5, 6].

Though previous research has explored placing air microphones near robot platforms to estimate liquid height [7] and pouring amounts [8], classify object materials [9] and categories [10, 11, 12], air microphones mainly capture sound waves transmitted through air, leading to noisy signals with ambient noises. On the other hand, contact microphones only sense the acoustic vibrations caused by physical contact. Past work has studied contact microphones for estimating the amount and flow of granular

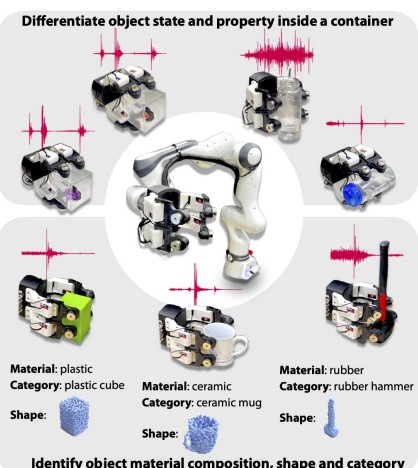

**Differentiate object state and property inside a container**

**Material**: plastic
**Category**: plastic cube
**Shape**:

**Material**: ceramic
**Category**: ceramic mug
**Shape**:

**Material**: rubber
**Category**: rubber hammer
**Shape**:

**Identify object material composition, shape and category**

Fig. 1: **SonicSense** enables container inventory status differentiation, heterogeneous material prediction, 3D shape reconstruction, and object re-identification on a diverse set of 83 real-world objects.

material [13], object position and category [14], and collectively performing object spatial reasoning for visual reconstruction [15].

Several major challenges remain to advance acoustic vibration sensing for robot object perception. Most current solutions focus on constrained settings with a small number ($N < 5$) of primitive objects [7, 8, 9, 13, 15, 16], homogeneous material composition for each object [9, 13, 16, 17],

8th Conference on Robot Learning (CoRL 2024), Munich, Germany.

single-finger testing [16, 17], and training and testing on different contacts but same objects [16, 17]. However, it is not clear whether such testing results can work with noisy and less controlled conditions. In addition, previous computational algorithms mainly utilize small machine learning models [10, 12, 16, 17] with a limited amount of data, which could be difficult to generalize. Moreover, the interaction mechanisms to collect acoustic data with objects rely on human manual movements [15, 16] or replaying pre-defined fixed robot poses [7, 8, 9, 10, 11, 13, 14, 16], making it difficult to scale to a large number of objects.

We present SonicSense (Fig. 1), a holistic design on both hardware and algorithm advancements for object perception through in-hand acoustic vibration sensing. Our design enables effective object perception abilities that are difficult for previous approaches to achieve altogether on 83 diverse real-world objects, including objects with complex geometry and heterogeneous materials. Our robot is capable of differentiating the inventory status of an occluded container through interactions. In more challenging tasks, through a naive but effective heuristic exploration policy to autonomously collect acoustic vibration characteristics, we can successfully infer material compositions, reconstruct the complete 3D object shape through sparse tapping, and re-identify previous objects. Additionally, we propose a set of end-to-end learning-based models by leveraging our large-scale dataset. Moreover, our design is cost-effective ($215.26) and easy to build with off-the-shelf components and 3D printing. Our experiments on large-scale real-world datasets demonstrate strong quantitative and qualitative performances of our approach. Overall, our method presents unique contributions and opens up new opportunities for robot tactile perception.

## 2  Related Works

**Tactile Perception** Tactile sensing [18, 19, 20] perceives object properties through contact. Vision-based tactile sensors [21, 22, 23, 24, 25] employ RGB cameras and light sources to detect robot skin deformations under contact. Barometric pressure tactile sensors [26, 27, 28, 29] perceive contact information by detecting the air or liquid pressure change between the soft elastomer and the sensor. Capacitive [30] and piezoresistive [31] sensors can monitor both static and dynamic pressure. Magnetic-based tactile sensors[32, 33, 34] achieve force detection by the movement of the magnetic field. Thermal conductivity and temperature tactile sensors[35] can identify the material and contact pressure by sensing the temperature change.

Our work explores the use of acoustic vibration as another modality for tactile perception. Acoustic vibration offers the advantage of detecting high-frequency vibration characteristics of objects using low power. Our design also features an intuitive mechanical and electrical design that is both affordable and robust. Furthermore, we will fully open-source our hardware, software, models, and dataset. Since it remains an open topic on which one type or combinations of certain types of tactile sensors serve as the optimal solution for tactile perception, we believe that it is valuable to explore different types of modalities and leave the investigation of combining them for future work.

**Acoustic Sensing for Object Perception** Extensive literature has investigated the leverage of acoustic sensing to estimate object states such as liquid amount [7, 8], object materials [9], object classes [10, 11, 12, 36], and object shapes and poses [15]. However, existing studies mostly focus on one or a few aspects of object states and have not been shown to be directly applicable to various aspects of object perception. In contrast, our method aims to generalize on multiple challenging object perception tasks ranging from solid and liquid object inventory status estimation to material classification, 3D shape reconstruction, and object re-identification.

Most recently, robot hand designs equipped with acoustic sensors have been demonstrated with strong performances on object material classification[16, 17], contact detection [16, 17, 37, 38], and object manipulations [39, 40, 41]. While recent efforts have achieved very promising results, current methods have focused on constrained settings with a small number of primitive objects [7, 8, 9, 13, 15, 16], homogeneous material composition [16, 9, 13], small machine learning models trained on a limited amount of data [16, 10, 12], and human manual data collection process by either moving the hand directly or replaying pre-defined poses with fixed contact point and forces [7, 8, 9, 10, 11, 13, 14, 15, 16]. Our work moves one step towards larger and more diverse set

of real-world objects with complex geometry and heterogeneous materials. With our simple but effective heuristic exploration policy, we can autonomously collect rich acoustic vibration responses of objects for training our learning-based computational models.

**Acoustic Sensing Dataset** To leverage the advancement of deep learning, there have been large efforts to create synthetic audio datasets [9, 42, 43]. Due to the intrinsic complexity of physical object properties, it remains challenging to close the simulation-to-real gap. For existing datasets created on real-world objects, the interaction is either performed by a human [3, 6, 43, 44, 45] or by an impact hammer[46] in a noise-controlled environment. Therefore, it is difficult to leverage such datasets for object perception in real-world robotics, whereas our work directly investigates acoustic vibration perception in real-world robotic systems.

## 3 The SonicSense

### 3.1 Robot Hand Design

Our robot hand (Fig. 2) has four fingers and each finger has one joint with one degree of freedom. Our intuitive mechanical design enables a range of interactive motion primitives for object perception including tapping, grasping, and shaking motions.

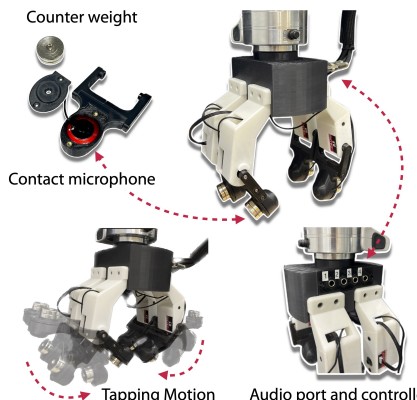

At each fingertip, a piezoelectric contact microphone is embedded inside the plastic shell while a round counterweight is mounted on the outer shell surface to increase the momentum of the finger motion. We found that the counterweight plays an important role in enabling large striking vibrations during tapping motion. Our contact microphones are synchronized and can pick up the acoustic vibrations at a frequency of $44,100$ Hz. Our design is highly affordable with a total estimated cost of $215.26 including all electronics components, motors, sensors, and 3D printing materials. See detailed splits and hardware specifications in Supp.A.

Fig. 2: Our robot hand includes four fingers where each fingertip is equipped with one contact microphone and a counterweight.

### 3.2 Real-World Object Dataset on Acoustics, Material, Shape, and Category

We have developed a dataset with 83 diverse real-world objects shown in Fig. 3. Our dataset covers nine material categories (i.e., plastic, glass, wood, metal, ceramic, paper, rubber, foam, and fabric) which includes challenging materials such as foam and fabric, and 22.9% of the objects include more than one material. We provide high-quality 3D scanned meshes and point cloud models for all objects, along with fine-grained per-point annotations of material categories on the point clouds. Our objects cover a variety of geometries, from simple primitives to complex shapes and from smaller objects to larger or longer objects. We also include objects that are traditionally difficult for vision sensors to perceive such as objects with transparent or reflective surfaces.

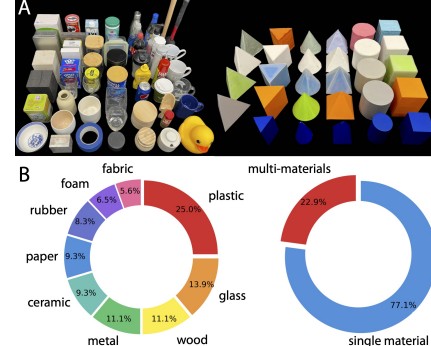

Fig. 3: (A) 83 real-world objects: 54 everyday objects and 29 3D-printed primitive objects with different materials attached to their surfaces. (B) The composition of the nine materials and multimaterial vs. single-material objects.

### 3.3 Heuristic-Based Interaction Policy

Unlike previous work that uses humans to manually hold the robot's hand to interact with objects or design fixed interaction poses and forces for replay, we derive a simple but effective heuristic-based interaction policy to collect the acoustic vibration response of objects. Our policy works well for all our real-world objects covering variable sizes and geometries. First, due to the unknown shape of the object, the policy will attempt to make contact with the object from both top-down and side directions, from high to low heights, with a fixed step size, until the first contact event is detected from each direction. Since our motor provides voltage resistance feedback, detecting contact events can be

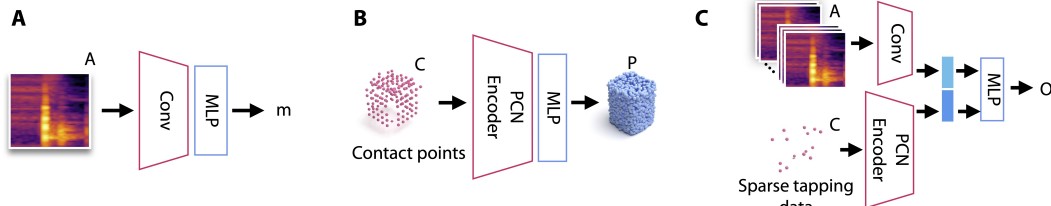

Fig. 4: **The network architectures** (A) Our material classification network takes in one Mel-spectrogram A from one tapping position through several convolutional and MLP layers and outputs the material label $m$. (B) Our shape reconstruction network takes in a set of sparse contact points $C$ through the Point Completion Network (PCN) encoder and MLP layers to output a dense and completed object point cloud $P$. (C) Our object re-identification model takes in both Mel-spectrogram A with 15 channels through several convolutional layers and their corresponding fifteen contact positions $C$ with PCN encoders. After fusing the features from the two networks through several MLP layers, our model outputs the final object label $O$.

achieved through voltage feedback or acoustic signal change. We used the voltage feedback due to its robustness and easy access. Second, from such initial exploration, we can estimate the height of the object (from the top-down direction) and its radius (from the side direction). Finally, with the estimated height and radius, the robot will use a grid sampling schedule to make sparse contact with the objects to collect acoustic responses. We provide a detailed description of our policy in Supp.B.

We make several assumptions when designing our interaction policy. First, we assume the maximum height of all objects. Second, we assume the size of the object can be held by the robot hand during tapping. Third, we fix the object on the table surface by following previous studies [47, 48, 49, 50]. Relaxing these assumptions may involve a learning-based exploration policy or dynamic tracking algorithm developments with tactile information which we leave as future work.

## 4    Object Perception from In-Hand Acoustic Vibration

### 4.1    Material Classification Model and Training

We aim to leverage in-hand acoustic vibration sensing on the task of material classification for *unknown objects*. Our material classification model takes in the Mel-spectrogram of the acoustic vibration signal from the impact sound and learns to predict the material label for that contact location. The network shown in Fig. 4(A) takes the form of three Convolutional Neural Network (CNN) layers followed by two Multilayer Perceptron (MLP) layers. Given a Mel-spectrogram $A_i$ of the i$^{th}$ sample in our interaction data, we train the material classification model $f_{mc}$, parameterized by $\theta_{mc}$, to output the corresponding material label category $\hat{m}_i$. We optimize the following loss function $\mathcal{L}_{mc}$ with the cross-entropy loss:

$$\min_{\theta_{mc}} \sum_i \mathcal{L}_{mc}(\hat{m}_i, m_i), \hat{m}_i = f_{mr}(A_i) \tag{1}$$

### 4.2    Shape Reconstruction Model and Training

Our shape reconstruction model takes the sparse contact points to generate a dense and complete 3D shape of the object. However, the contact point locations only serve as a rough estimation due to real-world interaction noises and ambiguity around the fingertips. Therefore, it is essential to leverage learning-based methods to capture the prior knowledge of 3D shapes of objects from training data. As shown in Fig. 4(B), We design the shape reconstruction model by referring to the Point Completion Network (PCN) [51]. We stack two PointNet [52] layers to encode the input contact points to a global feature vector with the dimension of $1 \times 1024$. We then feed the global feature vector into a decoder network with fully-connected layers to produce the final point cloud with $2,024$ points. Unlike the original PCN, we do not use the folding-based decoder.

Given a contact point cloud $C_i$ of the i$^{th}$ object collected through our robot hand interaction, our objective is to train the shape reconstruction model $g_{sr}$, parameterized by $\theta_{sr}$, to generate a dense and complete point cloud of the corresponding object $\hat{P}_i$. We optimize the objective function in

Eq. 2 with the Chamfer Distance (CD) loss in Eq. 3:

$$\min_{\theta_{sr}} \sum_i CD(\hat{P}_i, P_i), \hat{P}_i = g_{sr}(C_i) \tag{2}$$

$$CD(\hat{P}_i, P_i) = \frac{1}{|\hat{P}_i|} \sum_{x \in \hat{P}_i} \min_{y \in P_i} ||x - y||_2 + \frac{1}{|P_i|} \sum_{y \in P_i} \min_{x \in \hat{P}_i} ||y - x||_2 \tag{3}$$

Because of the challenging nature of this task and the limited amount of real interaction data, we constructed a simulation environment for such contact interaction to augment the dataset as shown in Fig. 5. We first pre-train the network with only our synthetic dataset to capture necessary prior knowledge and then gradually reduce the percentage of synthetic data and increase the percentage of real-world data during the training process. See Supp.E for the training schedules.

### 4.3 Object Re-identification Model and Training

When an object has been interacted with by the robot with its acoustic vibration responses, we aim to have our robot re-identify the object through a set of new tapping interactions. To make the problem harder, we restrict the new tapping interactions to happen on different locations on the object. In our object re-identification model, we input both the collection of Mel-spectrograms and their associated contact points to the network to predict the label of this object. As shown in Fig. 4(C), The model consists of three neural network components: an audio encoder to encode a set of spectrograms from acoustic vibration signals, a contact point encoder to encode a set of corresponding contact points, and a few layers of MLPs to fuse the encoded features to predict the object class among 82 objects.

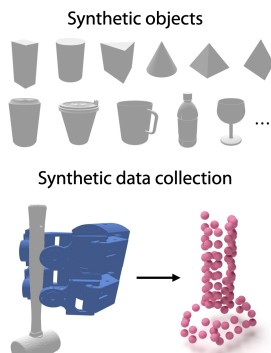

Synthetic objects

Synthetic data collection

Fig. 5: We conducted synthetic data collection of contact points on a large number of 3D objects in the simulation for data augmentation.

Given a set of spectrograms $A_i$ and contact point cloud $C_i$ of the i$^{\text{th}}$ object in our interaction data, we train the shape reconstruction model $h_{or}$, parameterized by $\theta_{or}$, to predict the corresponding object label $\hat{o}_i$. We optimize the following loss function $\mathcal{L}_{or}$ with the cross-entropy loss:

$$\min_{\theta_{or}} \sum_i \mathcal{L}_{or}(\hat{o}_i, o_i), \hat{o}_i = h_{or}(A_i, C_i) \tag{4}$$

### 4.4 Implementation Details

All models are trained on a single NVIDIA GeForce RTX 3090 GPU. All training lasts from less than an hour to a few hours depending on the tasks. We used grid search during training and selected the best set of hyperparameters based on the validation performance. We present the dataset details, model architecture, training hyperparameters, and detailed results in Supp.D-F. We will also open-source the hardware design, the software, the models, and the dataset.

## 5 Experimental Results

### 5.1 Characterizing Basic Sensing Capabilities of SonicSense

**Resistance Against Ambient Noise** To test whether SonicSense is robust against ambient noise, we compared the amplitude of signal recordings from SonicSense's fourth finger with the recording from an external air microphone. The overall setup is shown in Fig. 6 and the detailed information is in Supp.G. Starting from the natural environmental noise in a typical lab, we played Gaussian white noise through a speaker and gradually increased the noise decibels. As shown in Fig. 6, though the amplitude recorded from the air microphone increased by a few thousand units, the amplitude from our robot fingertip barely changed by a few units. This comparison suggests that our hardware design has a strong resistance against ambient noises and only focuses on the vibration signals through physical contact.

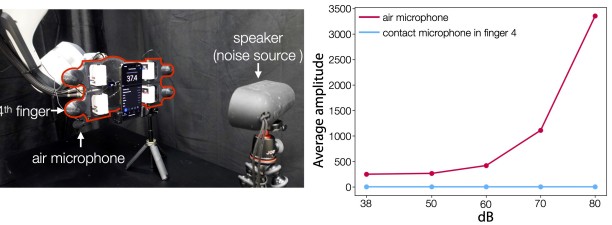

Fig. 6: **Test for resistance against ambient noise.** While the external air microphone sensor is highly sensitive to ambient noises, our robot does not capture any noticeable noises, making it focus on the vibration signals through physical contact.

**Differentiating Inventory Status in Containers** To assess whether SonicSense design can capture subtle but informative acoustic vibration signals to reveal object states in challenging scenarios, we conducted two experiments with a focus on rigid and liquid objects respectively.

We first placed different numbers of dice and then a series of dice with various shapes inside a plastic container. We had the robot hold the container and rotate it forward and backward by $180°$ around the wrist. In the second experiment, the robot held a bottle with three different initial amounts of water (e.i. 0ml, 100ml, and 200ml). We then poured $100\,\text{mL}$ water three separate times into the bottle with a constant flow using a dispenser bottle. Next, the robot held the bottle and performed a horizontal shaking motion with three amounts of water (i.e., $100\,\text{mL}$, $200\,\text{mL}$, and $300\,\text{mL}$).

From the visualization of the vibration signals captured by our robot in Fig. 7, we can tell that the signals, though subtle, indeed reflect the spatial and temporal features of different inventory statuses. Quantitatively, we derived twelve interpretable features based on traditional acoustic signal processing, including the root mean square of the signal, spectral centroid, bandwidth, contrast, flatness, roll-off, zero crossing rate, tempogram, poly features, Mel-frequency cepstral coefficients, chroma, and tonnetz, all averaged across time. We then performed an unsupervised nonlinear dimensionality reduction with t-SNE [53] on this 12-dimensional feature vector for all our experiments as shown in Fig. 7. The clear clusters indicate that SonicSense is able to provide informative cues to distinguish not only the numbers and geometries of solid objects but also the continuous and subtle liquid states in a small container.

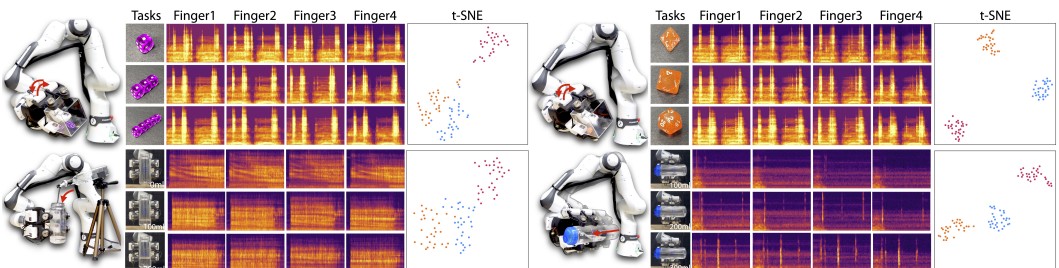

Fig. 7: **Four container experiments for characterizing the sensing capabilities of our hardware platform.** The Mel-spectrograms show one example of the collected acoustic vibration signal for qualitative visualizations. We repeated 30 trials for each object across all experiments. t-SNE results based on the twelve feature vectors are shown to visualize the difference in the acoustic vibration features between different sub-tasks represented by the three colors.

### 5.2 Material Classification

Unlike previous studies, in the material classification task, we split the data by objects to avoid overlapping acoustic events. We use a split of 60, 11, and 11 for training, validation, and testing. Since the testing dataset is unbalanced caused by different contact points for each object, we report the average F1 score as our evaluation metric. All our experiments are conducted with three different splits to obtain the mean and standard deviation.

As shown in Fig. 8(A), the initial result of our method leads to a $0.523$ F1 score. However, many errors stem from outlier-like predictions, even though most predictions in the surrounding object regions remain accurate. Therefore, we propose an iterative refinement procedure assuming that materials are relatively uniform and smooth around local regions. Our iterative algorithm works as follows: for each object, we first filter out the predictions with low occurrence with a threshold $M$

and reassign their labels with the highest occurrence label. For each point, we assign the label based on a majority vote among all its $K$ nearest neighbors. We then repeat this step for $N$ steps. The values of $M$, $K$, and $N$ are selected based on the best validation performance. We provide the full results in Supp.D. With this refinement algorithm, our final average F1 score reaches to $0.763$.

We compared our results with a random search baseline and a nearest neighbor baseline. For the nearest neighbor baseline, we compared each audio spectrogram in the test dataset with all spectrograms in the training dataset and selected the label based on the lowest mean square distance. Our method outperforms both baselines, suggesting that our algorithm generalizes beyond the training set and provides accurate material label prediction on *unseen objects*.

The confusion matrix in Fig. 8(B) indicates that our method can produce highly accurate predictions even on challenging soft surface materials such as foam and fabric, which typically do not create obvious striking signals.

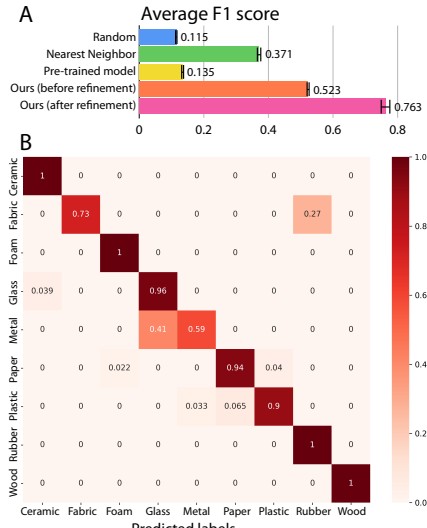

Fig. 8: (A) Quantitative results and baseline comparisons. (B) Confusion matrix.

This result implies that our method can capture subtle but critical signals through in-hand acoustic vibrations caused by the slight differences in the stiffness of these materials. Ceramics is usually confused with glass due to their similar properties. Predicting plastic material is challenging because the plastic materials in our dataset vary greatly in their thickness and stiffness.

Moreover, we experimented with pre-training our model with the recent audio-material dataset [6, 45, 54]. However, we found that this pre-training scheme hurts the performance (Fig. 8). The acoustic signals in these datasets were collected with air microphones and noise-controlled experiments. Hence, a large domain gap exists.

### 5.3 Shape Reconstruction

As shown in Fig. 9, through sparse contact tapping points, we obtained an average of $0.00876$m Chamfer-L1 distance score. Fig. 10 shows examples of our shape reconstruction results on our testing dataset. We compared our results with a random search baseline and a nearest neighbor baseline. For the nearest neighbor baseline, we compared our test tapping points with all tapping points in our training dataset and selected the corresponding ground-truth point cloud with the lowest Chamfer-L1 distance as the prediction. The Chamfer-L1 distances for these baselines are $0.03653$m and $0.01347$m respectively. Our model greatly outperforms both baselines and shows strong generalization abilities on unknown shape reconstructions.

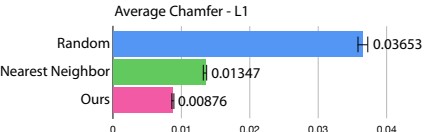

Fig. 9: Chamfer-L1 distance between the reconstructed point cloud and the ground-truth point cloud.

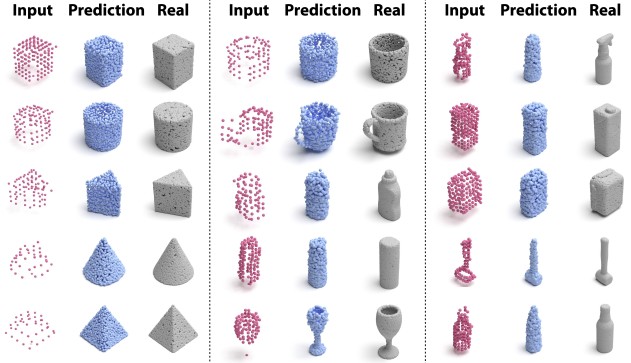

Fig. 10: SonicSense can produce a complete and accurate 3D point cloud of objects from sparse, nonuniform, and noisy contact positions.

The prediction on objects with primitive shapes generally has near perfect performance. Additionally, our method exhibits the capability to reconstruct objects with concave geometries. It is also worth noting that there is only one wine glass, as shown in the middle of column three, in our real

object dataset. Therefore, this result is likely achieved through our augmentation dataset in the simulation. Some failure prediction examples include the nozzle of the spray and the cap of the bottle, due to the limited number of spray objects in our dataset and its complex shape within a small region.

### 5.4 Object Re-identification

We evaluated our method with fifteen channels of spectrograms from fifteen random contact locations, which is analogous to our robot random tapping objects a few times. Our model can accurately re-identify the objects with a 92.52% test accuracy while the random baseline and nearest neighbor baseline only gave 1.33% and 43.45% respectively, as shown in Fig. 11. By looking into the detailed confusion matrix of our testing results as provided in the supplementary files, we can see that smaller

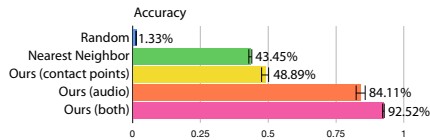

Fig. 11: SonicSense can accurately re-identify the objects that it has interacted with while sensing their acoustic vibration responses.

objects are more difficult due to the limited number of interaction data. We can also observe that the model performs worse when the objects are relatively small, and the materials have similar acoustic properties, such as ceramic and glass.

Additionally, in order to verify the importance of both the shape and material information for this object re-identification task, we conducted two ablation studies to either remove the acoustic vibration input or the tapping point input. The acoustic vibration-only network reaches an accuracy of 84.11%, and the tapping point-only network reaches an accuracy of 48.89%. Therefore, acoustic information provides a more informative representation of the object and the performance will be further improved by incorporating an additional modality of rough contact positions for object re-identification.

## 6 Conclusion

We have introduced SonicSense, an integrated hardware and software solution to enable rich object perception capabilities with in-hand acoustic vibration for a multi-finger robot hand. Our experimental results demonstrate the versatility and efficacy of our design on varieties of object perception tasks including solid and liquid object inventory status estimation within containers, material classification, 3D shape reconstruction, and object re-recognition. Unlike previous approaches, our study involves a significantly larger number of real-world objects with complex geometry and heterogeneous materials. Our testing is also conducted on unseen objects instead of different contact events on the same objects. Our simple but effective exploration policy avoids previous manual interaction or fixed pose replay. Our investigations outline the challenges and necessities of considering real-world noises and robot-specific interactions for robot object perception. Overall, our method presents unique contributions to tactile perception with acoustic vibrations and opens up new opportunities for future robot designs to build a more robust, complete, versatile, and holistic perceptual model of the world.

**Limitations and Future Work** We see a number of opportunities to improve our current approach in future research. One immediate assumption to relax is to avoid fixing the objects on the table. Existing previous work has not been able to relax such assumptions. However, recent efforts on object tracking with tactile information [55] are promising. Future work can consider adapting such algorithms as an online object estimation and tracking within the interaction policy. Our work does not consider multiple objects. However, in the real world, acoustic vibrations can also be helpful for object perception in cluttered environments. Moreover, our current hardware design does not consider complex robot manipulation skills such as dexterous manipulations. With a more anthropomorphic and higher degree-of-freedom hand design, embedding our findings of in-hand acoustic vibration sensing can enable future exploration of dexterous interactions with everyday objects. Furthermore, though our focus in this paper is acoustic vibration sensing for object perception, future work shall involve integrating multiple sensing modalities to obtain more complementary information for robots to work in complex and unstructured environments.

**Acknowledgments**

This work is supported by ARL STRONG program under awards W911NF2320182 and W911NF2220113, by DARPA FoundSci program under award HR00112490372, and DARPA TIA-MAT program under award HR00112490419.

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

# Supplementary Material

## A. Hardware Configuration of Acoustic Robot Hand

Our hardware is built by 3D printing with Polylactic Acid (PLA) filament. We use the LX-224 servo motor to actuate the finger which provides 20kg·cm torque and accurate position and voltage feedback. The contact microphones we used are commercially available on Amazon and the two counter weights on each fingertip is 40g. We place the controller of the motors as well as the audio jack inside the palm. The cost split of building our acoustic robot hand is shown in Tab. 1.

| Part | Amount | Price in total ($) |
|------|--------|--------------------|
| Lx224 motor | 4 | 75.96 |
| TTL/USB Debugging Board | 1 | 12.99 |
| 4pcs piezo contact microphone | 1 | 28.58 |
| Audio cable | 4 | 59.96 |
| Counterweight | 8 | 27.44 |
| PLA 3D printing material (689.11g) | 1 | 10.33 |
|  | Total: | 215.26 |

Tab. 1: **Cost of the acoustic robot hand.**

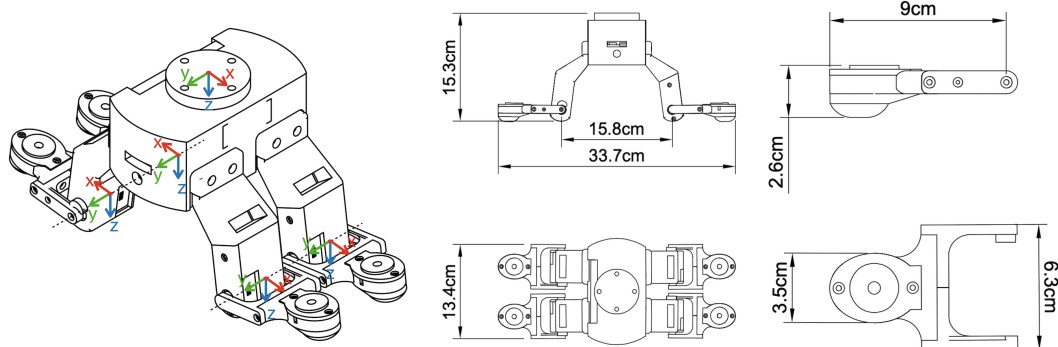

Fig. 12: **CAD model and coordinate system of the robot hand.**

**Contact Position Calculation**

We will introduce the calculation of the approximated contact point location in this section. We set the fingertip position at the center of the fingerprint. Once the binary contact event is detected, we will record the angle of each finger joint. Based on our CAD model, we can also obtain other kinematic parameters of the hand such as the length of each link. Fig. 12 shows the sketch of our hardware design and its coordinate system. We use the center of the palm as the origin of the hand. The y-axis is parallel to the four finger joints and the z-axis is perpendicular to the palm, facing towards the finger. The coordinate system of the finger joints has the same orientation as the above central coordinate system of the hand. The center of the finger joint coordinate is located at the center of each motor. We denote the position of joint 1 as $p_1 = \begin{bmatrix} x_1 \\ y_1 \\ z_1 \end{bmatrix}$ , where $x_1 = 7.845cm$, $y_1 = 3.429cm$, $z_1 = 13.691cm$. Considering that the four fingers are symmetrical, the positions of the rest of the fingers can be represented as

$$p_2 = \begin{bmatrix} x_1 \\ -y_1 \\ z_1 \end{bmatrix}, p_3 = \begin{bmatrix} -x_1 \\ y_1 \\ z_1 \end{bmatrix}, p_4 = \begin{bmatrix} -x_1 \\ -y_1 \\ z_1 \end{bmatrix} \tag{5}$$

We use $Pe_i$ to represent the fingertip position under its joint coordinate. Since the length between the origin of joint coordinate and the fingertip is $l = 7.6cm$, the fingertip position of finger $i$ under

the robot hand coordinate, denoted as $^{H}Pe_i$, can be calculated by

$$^{H}Pe_i = p_i + RPe_i \tag{6}$$

Here, $Pe_i = \begin{bmatrix} l \\ 0 \\ 0 \end{bmatrix}$. For finger 1 and finger 2, $R$ can be represented as:

$$R = R_y(-\theta_i + 4.5^{\circ}) \tag{7}$$

The $\theta_i$ here represents the joint angle of finger $i$. For finger 3 and finger 4, $R$ can be represented as:

$$R = R_y(\theta_i - 4.5^{\circ})R_z(180^{\circ}) \tag{8}$$

We attach the robot palm on the end effector of the Franka Emika Panda arm and align the end effector coordinate with the robot hand coordinate to ensure $^{H}Pe_i = {}^{E}Pe_i$. Having the transformation matrix of the end effector coordinate to the robot arm $^{R}_{E}T$, we can then calculate the contact position under the robot arm coordinate, represented as $^{R}Pe_i$, by:

$$^{R}Pe_i = {}^{R}_{E}T\,{}^{E}Pe_i \tag{9}$$

## B. Interaction Policy for Data Collection

We will introduce the interaction policy for the robot to conduct real-world data collection in this section. Since vision is not used, the dimension of the object is unknown. Therefore, the first challenge of the interaction policy is to estimate the dimension of the object.

We fix the object at the center of a black base on top of a wooden board as shown in Fig. 13A. The robot will first perform two tapping motions from the side with two different heights (Fig. 13B and 13C) and save the valid contact points. The tapping motion is able to make contact with objects since the objects' body is across the centerline of the black base. Based on the highest detected contact point location, the robot will have a rough estimation of the height of the object.

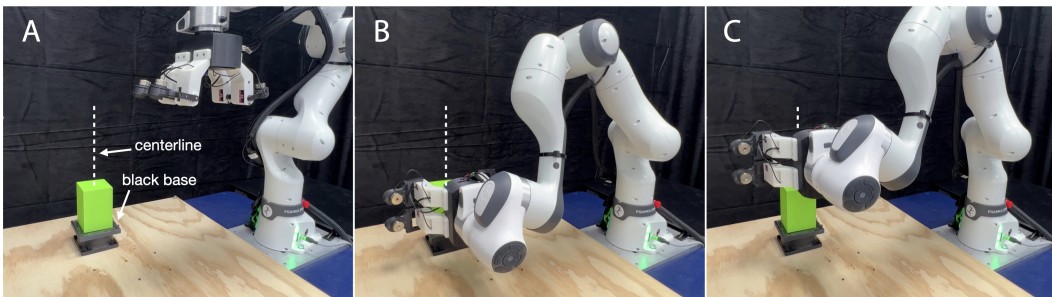

Fig. 13: **Initial estimation stage of the interaction policy.** (**A**)The black base and its centerline. (**B**)The first side tapping in a lower position. (**C**)The second side tapping in a higher position.

To obtain a more accurate estimation of height and radius of the object, the robot will reset to its initial home position to lift its two fingers in front, and use its third finger to explore the highest edge on top of the object. Specifically, starting from a suitable height away from the objects, the third finger of the hand moves towards the centerline of the black base (Fig. 14A) and reaches another side of the centerline (Fig. 14B). The robot keeps monitoring the acoustic signal from the third finger during this process. Once a contact event is detected, the robot will record the contact position through the kinematic model and stop exploring. If the third finger does not make contact with the object and there is no contact event being detected, the robot hand will move back to the initial position, and move down a little bit as shown in Fig. 14C and keep executing the same edge exploring motion until a contact sound is detected as shown in Fig. 14D. We use the distance between

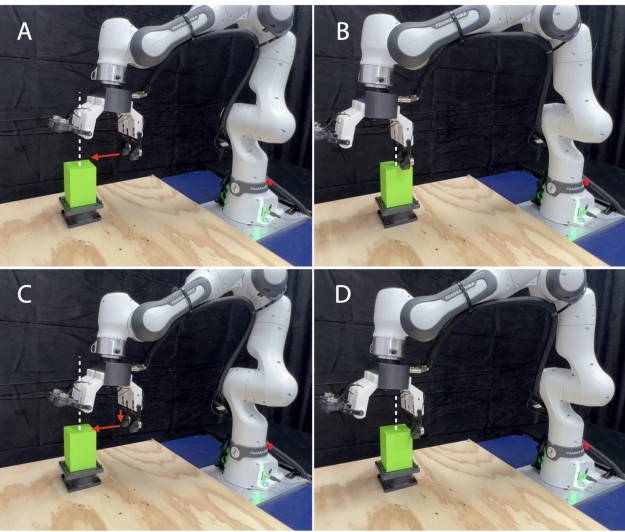

Fig. 14: **Exploration of highest edge.** (**A**)The third finger gets to a position higher than the roughly estimated height and moves toward the centerline of the black base. (**B**)The third finger reaches the centerline and no contact happens. (**C**) The third finger goes back to the previous initial position as shown in (**A**), moves down a little bit, and moves toward the centerline again. (**D**)The third finger makes contact with the object. A contact event is detected and the contact position is saved.

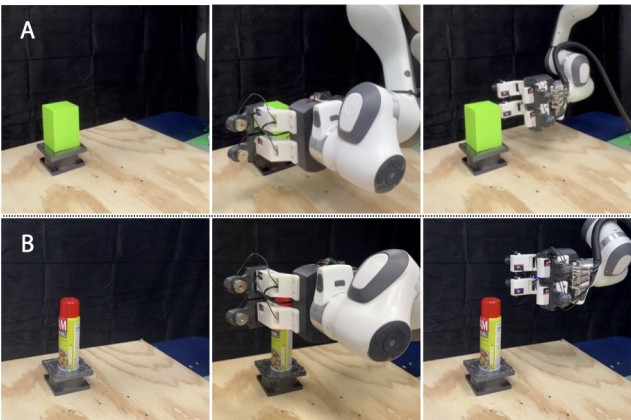

Fig. 15: **Two scales of left side and back side tapping.** (**A**) The lower starting position of these two tapping motions on a smaller object. (**B**)The higher starting position of these two tapping motions on a larger object.

the contact point and the centerline as the estimated radius of the object and the distance between the contact point and the black base plane as the final estimated height of the object.

Following the estimation, the robot will interact with the object through tapping motions on three sides of the object in sequences including the top side, the left side, and the backside of the object. During each tapping sequence, the robot hand will approach and tap the object step by step, ensuring continuous tapping while gradually transitioning from top to bottom (or left to right for top tapping) to cover the entire surface of the object. A detailed illustration of the tapping motion is provided in Fig. 16. The robot relies on the estimated radius to determine whether to collect tapping data on top of the object since the useful information is limited when the surface area on top of the object is too small. We set the radius threshold to be 2cm in this case. Additionally, depending on the estimated height of the objects, the robot decides to start from a higher position or a lower position for two different scales of data collection during the left-side tapping and back-side tapping

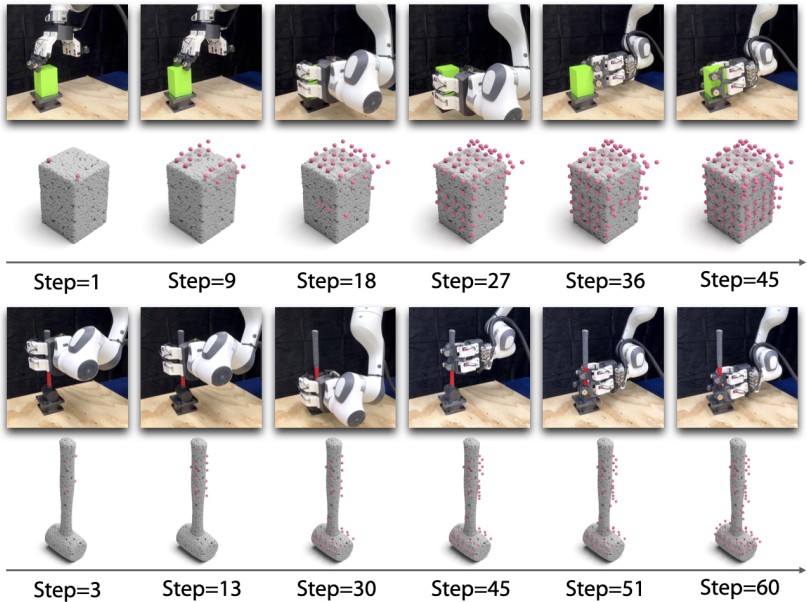

Fig. 16: **Visualizations of contact events through tapping motion.** We show the sequence of tapping interactions on a smaller object in the first row and a larger object in the second row. The x-axis shows the step progression. The estimated contact fingertip positions are shown as points overlaid on top of the ground-truth 3D shapes.

as illustrated in Fig. 15. We set the height threshold to be 20cm. For a more comprehensive visual demonstration, please refer to our Supplementary Movies. When our robot hand taps objects, we can determine such contact events from the voltage feedback from the motor in each finger. We put a 10Ω resistor in series with the servo motor power. When the motor encounters external force, it receives an increase of current supply and, as a result, a decrease of voltage from the motor voltage sensor readings. Though we also experimented with detecting salient acoustic signals to detect the binary contact event, we found that the natural motor voltage feedback embedded by the motors provides more reliable and accurate signals. This is because the voltage feedback can reflect subtle resistance encountered by the robot finger, where acoustic vibrations from the contact microphone also include motor noises. After the contact event is detected, we can localize the contact points in 3D space with respect to the robot with the forward kinematics of our robot hand and arm.

We conducted this real-world data collection on the 83 real-world objects mentioned in our main texts and used the tapping data only with valid contact events for our material classification, shape reconstruction, and object re-recognition tasks. The tapping data are saved in txt file format, including position in the x-axis ($x$), position in the y-axis ($y$), position in the z-axis ($z$), binary tapping detection from acoustic vibration signal (a), binary tapping detection from voltage signal (v), finger index (f), and index of tapping (i):

$$\{x, y, z, a, v, f, i\} \tag{10}$$

Each tapping data is also accompanied by a five-second audio recording, which is guaranteed to cover the acoustic signals of the impact sound. We will present the audio processing procedure in the next section.

## C. Acoustic Vibration Recording and Processing

To obtain more useful information, we first extracted the striking vibration signal from the 5-second recording. Typically, the anticipated tapping vibration signal exhibits a noticeable characteristic, often resembling a peak-shaped pattern. However, real-world acoustic vibration signals may contain

random noises, so simply extracting the signal around maximum amplitude will not work. To extract informative acoustic vibration signals, we used a window of 1000 units of acoustic waveform to monitor the average absolute amplitude of the acoustic signals. If the average amplitude in the current window is larger than both the average amplitude in the previous window and the next window, 20000 units of the acoustic signals ($20000/44100Hz = 0.45351474s = 453.5147ms$) will be extracted around the beginning timestamp of the current window. We applied the same extracting mechanism to the recording with no obvious striking signal, for example, foam objects. For those objects, most extracted signals are motor noises. However, we consider this as a feature of the soft materials perceived by our robot hand. Finally, we followed previous work to convert the raw audio signals to Mel spectrogram representations with a dimension of $64 \times 64$ resolution. The length of the Fast Fourier transform (FFT) window is set to 2048 and the number of the Mel bands is set to 64. The highest frequency is restricted to 8192Hz. To ensure the correct dimension of the Mel spectrogram along the temporal dimension, the number of samples between consecutive frames is rounded to the nearest value obtained by dividing the total number of audio samples by 64.

## D. Implementation Details for Material Classification

**Dataset Construction**

In this section, we explain how we obtained the material label for each of the contact positions around the object. During tapping data collection, we used two depth cameras to capture two point clouds of the object after fixing the object on the platform. We fuse those two point clouds and use the annotated point cloud model to align with the object. We then searched the nearest point of the annotated point cloud for each of the contact points and assigned the annotated material label to that contact point based on the label of the nearest point. The material label serves as the ground truth for training our material classification model.

Tapping data from 82 objects is used for the material classification task. We removed the acoustic data collected from a disinfecting wipe because the paper material is too soft and there is a thin plastic film attached to the paper, so the signal is too unique and out of distribution from all other paper materials. We conducted separate training on three different random splits of the dataset to evaluate our method. Note that our splits are based on objects to avoid strong overlapping between training, validation, and testing data. We have 60 objects for training, 11 objects for validation, and 11 objects for testing. We balanced the number of acoustic data in each of the material classes for training and validation by duplicating them from random choice. Therefore, due to different objects coming with different sizes and different numbers of tapping data, the number of data points for each split will be different under the same number of objects. The resulting number of data points in each split is shown in Tab. 2.

| Dataset splits | Training | Validation | Testing |
|---|---|---|---|
| Split 1 | 8829 | 1521 | 948 |
| Split 2 | 7767 | 2142 | 1293 |
| Split 3 | 8433 | 1278 | 1195 |

Tab. 2: **Material classification data statistics.**

**Material Classification Model Architecture**

We show the detailed architecture design of our material classification network in Tab. 3.

**Material Classification Hyperparameters and Results**

We list all the hyperparameter selections and experimental results across different random data splits in Tab. 4. We select the model based on the best validation accuracy. Due to the testing dataset being

| Layer name | Input Channel | Output Channel | Kernal Size | Stride Size | Padding |
|---|---|---|---|---|---|
| Conv1 | 1 | 16 | 6 | 2 | 0 |
| Batch norm + Relu | N/A | N/A | N/A | N/A | N/A |
| Maxpool1 | 16 | 16 | 2 | 2 | 0 |
| Conv2 | 16 | 32 | 5 | 1 | 0 |
| Batch norm + Dropout+ Relu | N/A | N/A | N/A | N/A | N/A |
| Maxpool2 | 32 | 32 | 2 | 2 | 0 |
| Conv3 | 32 | 150 | 5 | 1 | 0 |
| Batch norm + relu+dropout | N/A | N/A | N/A | N/A | N/A |
| Fc1 | 150 | 70 | N/A | N/A | N/A |
| Dropout | N/A | N/A | N/A | N/A | N/A |
| Fc2 | 70 | 9 | N/A | N/A | N/A |

Tab. 3: **Neural network architecture of the material classification model.**

unbalanced, we use the average F1 score as our evaluation metric. The confusion matrix of the first experiment is shown in Fig. 17.

| | Split 1 | Split 2 | Split 3 |
|---|---|---|---|
| Max epoch | 300 | 300 | 300 |
| Learning rate | 0.00903 | 0.00935 | 0.00903 |
| Dropout | 0.52105 | 0.40947 | 0.30927 |
| Batch size | 36 | 32 | 32 |
| Optimizer | | SGD | |
| LR schedule | | Decays the Learning rate by 0.1 every 200 steps | |
| Best train accuracy | 0.9891 | 0.9876 | 0.9826 |
| Best validation accuracy | 0.6141 | 0.5691 | 0.5696 |
| Refinement parameters (M,K,N) | (8,3,25) | (8,8,25) | (6,1,30) |
| Testing F1 score | 0.5711 | 0.5035 | 0.4939 |
| Testing F1 score after refinement | 0.8786 | 0.6924 | 0.718 |

Tab. 4: **Material classification hyperparameters and results on three random data splits.**

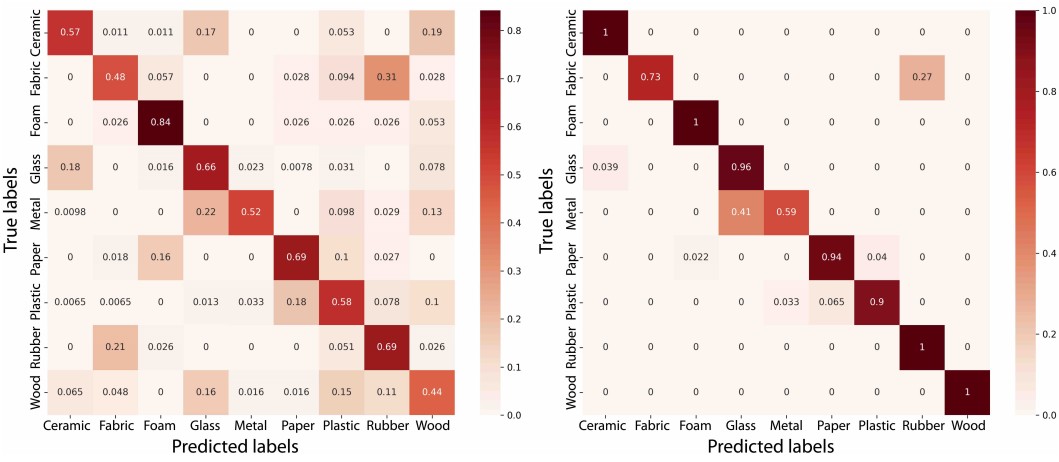

Fig. 17: **Confusion matrix of material classification task on one testing dataset** The results of the initial prediction(left) and the results of the prediction after refinement(right).

## E. Implementation Details for Shape Reconstruction

### Dataset Construction

We leverage the contact positions collected from the 83 objects for our shape reconstruction task. The tapping interaction results in highly sparse contact points. The maximum number of contact points is 300 for each object. We divided the dataset based on 15 shape categories(i.e., bottle, can cup, hammer, mug, wine glass, cube, cube(concave), cylinder, cylinder(concave), cone, quadrangular pyramid, triangular pyramid, prism, irregular) and split the dataset for training, validation, and

testing according to these categories. To balance the training dataset for learning, we randomly duplicated the data up to 100 for each category. We then randomly sampled 80% to 90% points and augmented the dataset to 1500 for each shape to augment the training dataset.

To further boost the training dataset, we leveraged the PyBullet simulator to collect synthetic tapping points. We collected 629 synthetic 3D models under the same categories of real-world objects. In the simulation, We implemented the same interaction policy and data augmentation techniques as in our real-world data construction. The ground truth model consists of a point cloud model comprising 5000 points for both the real-world objects and the synthetic objects. Our experiments include three different splits of data with different random seeds where both the validation objects and testing objects only include real-world objects. For real-world objects, we split them into 61, 11, and 11 objects for training, validation, and testing. In order to leverage the simulation data to improve our model on our real-world objects, initially, we use all the synthetic datasets to pre-train our model. Gradually, we blended in more real-world data and reduced the percentage of synthetic data in our training set. We show the blending schedule in Tab. 5.

| Epoch | Synthetic dataset | Real dataset |
|---|---|---|
| 0-100 | 100% | 0% |
| 100-200 | 90% | 10% |
| 200-300 | 80% | 20% |
| 300-400 | 60% | 40% |
| 400-500 | 40% | 60% |
| 500-600 | 20% | 80% |
| 600-700 | 10% | 90% |
| 700-800 | 5% | 95% |
| 800-1000 | 0% | 100% |

Tab. 5: **The blending schedule of synthetic dataset and real dataset during shape reconstruction training.**

## Shape Reconstruction Model Architecture

We show the detailed architecture design of our shape reconstruction network in Tab. 6.

| Layer name | Input Channel | Output Channel | Kernal Size | Stride Size | Padding |
|---|---|---|---|---|---|
| Conv1d | 3 | 128 | 1 | 1 | 0 |
| Batch norm + Relu | N/A | N/A | N/A | N/A | N/A |
| Conv1d | 128 | 256 | 1 | 1 | 0 |
| Conv1d | 512 | 512 | 1 | 1 | 0 |
| Batch norm + Relu | N/A | N/A | N/A | N/A | N/A |
| Conv1d | 512 | 1024 | 1 | 1 | 0 |
| Fc1 | 1024 | 1024 | N/A | N/A | N/A |
| Relu | N/A | N/A | N/A | N/A | N/A |
| Fc2 | 1024 | 1024 | N/A | N/A | N/A |
| Relu | N/A | N/A | N/A | N/A | N/A |
| Fc3 | 1024 | $3 \times 2000$ | N/A | N/A | N/A |

Tab. 6: **Neural network architecture of the shape reconstruction model.**

## Shape Reconstruction Hyperparameters and Results

We list all the hyperparameter selections and experimental results across different random data splits in Tab. 7.

| | Split 1 | Split 2 | Split 3 |
|---|---|---|---|
| Max epoch | | 1000 | |
| Learning rate | | 0.000005 | |
| Batch size | | 500 | |
| Optimizer | | Adam | |
| LR schedule | | Decays the Learning rate by 0.7 every 500 steps | |
| Lowest validation loss(CD-L2) | 3.6634e-05 | 5.6421e-05 | 5.4612e-05 |
| Lowest testing loss(CD-L2) | 6.7536e-05 | 5.6938e-05 | 6.0342e-05 |
| Lowest testing loss(CD-L1) | 0.009184 | 0.0085316 | 0.0085615 |

Tab. 7: **Shape reconstruction hyperparameters and results on three random data splits.**

# F. Implementation Details for Object Re-recognition

## Dataset Construction

We utilize both the tapping vibration signal and contact points from all 82 objects for our object re-recognition task. For each of the objects, we first randomly split 20% of the tapping data for testing, 20% of the tapping data for validation, and 60 % of the tapping data for training. To augment the dataset, 15 tapping data are randomly sampled 500 times for each object in the training dataset, and 50 times for each object in the validation and testing dataset. Therefore, there are 410,000 data points for training, 4,100 data points for validation, and 4,100 data points for testing. As in the previous two tasks, we have three different random splits of the dataset for evaluation.

## Object Re-recognition Model Architecture

We show the detailed architecture design of our object re-recognition network in Tab. 8.

| Layer name | Input Channel | Output Channel | Kernal Size | Stride | Padding |
|---|---|---|---|---|---|
| **Audio encoder:** | | | | | |
| Conv1 | 15 | 16 | 6 | 2 | 0 |
| Batch norm + Relu | N/A | N/A | N/A | N/A | N/A |
| Maxpool1 | 16 | 16 | 2 | 2 | 0 |
| Conv2 | 16 | 32 | 5 | 1 | 0 |
| Batch norm + Dropout+ Relu | N/A | N/A | N/A | N/A | N/A |
| Maxpool2 | 32 | 32 | 2 | 2 | 0 |
| Conv3 | 32 | 150 | 5 | 1 | 0 |
| Batch norm + relu | N/A | N/A | N/A | N/A | N/A |
| **Contact point encoder:** | | | | | |
| Conv1d | 3 | 64 | 1 | 1 | 0 |
| Batch norm + Relu | N/A | N/A | N/A | N/A | N/A |
| Conv1d | 64 | 64 | 1 | 1 | 0 |
| Conv1d | 128 | 128 | 1 | 1 | 0 |
| Batch norm + Relu | N/A | N/A | N/A | N/A | N/A |
| Conv1d | 128 | 150 | 1 | 1 | 0 |
| **MLP layers:** | | | | | |
| dropout | N/A | N/A | N/A | N/A | N/A |
| fc1 | 300 | 170 | N/A | N/A | N/A |
| dropout | N/A | N/A | N/A | N/A | N/A |
| fc2 | 170 | 170 | N/A | N/A | N/A |
| dropout | N/A | N/A | N/A | N/A | N/A |
| fc3 | 170 | 82 | N/A | N/A | N/A |

Tab. 8: **Object re-recognition model**

## Object Re-recognition Hyperparameters and Results

We list all the hyperparameter selections and experimental results across different random data splits in Tab. 9.

| | Split 1 | | | Split 2 | | | Split 3 | | |
|---|---|---|---|---|---|---|---|---|---|
| Max epoch | 500 | | | 500 | | | 500 | | |
| Learning rate | 3.6515e-05 | | | 8.8224e-05 | | | 7.8910e-05 | | |
| Dropout | 0.2348 | | | 0.2240 | | | 0.3253 | | |
| Batch size | 200 | | | 200 | | | 400 | | |
| Results with different input: | **A+C** | **A** | **C** | **A+C** | **A** | **C** | **A+C** | **A** | **C** |
| Best validation accuracy | 0.9707 | 0.8744 | 0.5295 | 0.9383 | 0.8756 | 0.5105 | 0.9200 | 0.8137 | 0.5456 |
| Best test accuracy | 0.9241 | 0.8034 | 0.4599 | 0.9183 | 0.8281 | 0.5276 | 0.9332 | 0.8920 | 0.4793 |

Tab. 9: **Object re-recognition hyperparameters and results on three random data splits.** In the table, "A" refers to the acoustic vibration signal input modality, and "C" refers to the contact point location input modality.

## G. Implementation Details for Resistance Test Against Ambient Noise

We placed a Saramonic LavMicro U1A air microphone next to the fourth finger. We utilized the National Institute for Occupational Safety and Health (NIOSH) Sound Level Meter App on an iPhone to monitor the noise level. This App has been shown to effectively capture accurate noise levels [56]. We stretched the finger so that we can place the iPhone microphone, the fingertip surface, and the air microphone in the same plane. We then set up a speaker facing toward the middle of the robot hand to play Gaussian white noise.

## H. Parameters of Acoustic Vibration Signal Processing in Characterization Experiments

We list all the parameters used to process the acoustic vibration signals for our sensing characterization experiments in Tab. 10. The parameters are used to convert the waveform representation into spectrogram representation as well as the twelve descriptor extraction methods. For liquid object experiments (i.e., shaking and pouring), the duration of the acoustic clip is long, so we chose a larger hop length. We have observed that there are no obvious acoustic signals above the frequency 8192 for this experiment, we chose the smaller highest frequency to show more details of the spectrograms. For rigid object experiments (i.e., dice shape and dice inventory), the duration of collision vibration is very short. To perceive more details of the collision signal, we chose a smaller hop length. Since the collision vibration signal of solid objects includes a higher frequency, we set the highest frequency to be larger.

| parameters | pouring | shaking | dice shape | dice inventory |
|---|---|---|---|---|
| Length of FFT window | 2048 | 2048 | 2048 | 2048 |
| Hop length | 2048 | 1024 | 512 | 512 |
| Number of Mel band | 64 | 64 | 64 | 64 |
| Highest frequency(Hz) | 8192 | 8192 | 16384 | 16384 |
| Color limits | [-10,-80] | [-10,-80] | [-10,-80] | [-10,-80] |
| Duration s | 13.00s | 2.38s | 3.89s | 3.89s |
| Poly features order | 3 | 3 | 3 | 3 |
| Roll-off percentage | 0.9 | 0.9 | 0.9 | 0.9 |

Tab. 10: **Parameters of acoustic vibration signal processing in characterization experiments.**

## I. Quantitative Results in Characterization Experiments

We derived twelve interpretable signal descriptors to quantitatively measure the features of the acoustic vibration signals. These feature descriptors are key statistical summaries of certain aspects of the signals, which provide an interpretable understanding of the signal-capturing capabilities and sensitivity characterization of our robot hand. Specifically, we denote them as (1) D1: average root mean square of the signal, (2) D2: average spectral centroid, (3) D3: average bandwidth, (4) D4:

average contrast, (5) D5: average flatness, (6) D6: average roll-off, (7) D7: average zero crossing rate, (8) D8: average tempogram, (9) D9: average poly features, (10) D10: average MFCCs, (11) D11: average chroma, and (12) D12: average tonnetz.

We repeated 30 trials for each subtask in the container experiment. In each trial, we extracted the twelve descriptor features by averaging the feature of the acoustic signals from four fingers. We then rescaled all the descriptor values to the range between 0 and 1. With these feature descriptors, we performed unsupervised dimensionality reduction of the high-dimensional signals into 2D space to test whether we can distinguish between various events triggered by different object states. The quantitative results of the mean value and standard error of the mean (SEM) of the twelve acoustic vibration signal descriptors from the 30 trials are shown in Tab. 11.

| | D1 | D2 | D3 | D4 | D5 | D6 |
|---|---|---|---|---|---|---|
| Dice(quantity:1) | 0.093±0.010 | 0.249±0.013 | 0.514±0.032 | 0.464±0.036 | 0.182±0.013 | 0.430±0.023 |
| Dice(quantity:3) | 0.590±0.011 | 0.825±0.016 | 0.827±0.019 | 0.3386±0.024 | 0.782±0.022 | 0.848±0.015 |
| Dice(quantity:5) | 0.803±0.013 | 0.779±0.023 | 0.728±0.034 | 0.306±0.039 | 0.684±0.034 | 0.798±0.026 |
| Dice(6 edges) | 0.106±0.009 | 0.839±0.020 | 0.886±0.009 | 0.482±0.027 | 0.613±0.022 | 0.823±0.016 |
| Dice(12 edges) | 0.559±0.011 | 0.682±0.015 | 0.655±0.011 | 0.517±0.035 | 0.717±0.027 | 0.615±0.010 |
| Dice(30 edges) | 0.802±0.014 | 0.415±0.027 | 0.299±0.023 | 0.695±0.029 | 0.469±0.036 | 0.388±0.024 |
| Pouring(1st 100ml) | 0.507±0.022 | 0.386±0.024 | 0.311±0.025 | 0.684±0.033 | 0.360±0.037 | 0.335±0.024 |
| Pouring(2nd 100ml) | 0.821±0.019 | 0.135±0.0137 | 0.154±0.015 | 0.796±0.018 | 0.326±0.033 | 0.114±0.010 |
| Pouring(3rd 100ml) | 0.751±0.016 | 0.163±0.0164 | 0.220±0.16 | 0.508±0.021 | 0.400±0.038 | 0.130±0.011 |
| Shaking(100ml) | 0.032±0.004 | 0.825±0.020 | 0.853±0.011 | 0.573±0.052 | 0.597±0.033 | 0.919±0.011 |
| Shaking(200ml) | 0.219±0.017 | 0.736±0.018 | 0.657±0.012 | 0.582±0.023 | 0.566±0.023 | 0.713±0.011 |
| Shaking(300ml) | 0.480±0.039 | 0.318±0.031 | 0.297±0.026 | 0.565±0.029 | 0.231±0.026 | 0.302±0.027 |
| | D7 | D8 | D9 | D10 | D11 | D12 |
| Dice(quantity:1) | 0.123±0.019 | 0.161±0.019 | 0.135±0.012 | 0.284±0.031 | 0.298±0.025 | 0.464±0.048 |
| Dice(quantity:3) | 0.739±0.022 | 0.524±0.021 | 0.576±0.018 | 0.281±0.023 | 0.716±0.020 | 0.553±0.042 |
| Dice(quantity:5) | 0.767±0.025 | 0.716±0.026 | 0.782±0.022 | 0.566±0.027 | 0.742±0.028 | 0.587±0.035 |
| Dice(6 edges) | 0.524±0.036 | 0.576±0.031 | 0.062±0.006 | 0.188±0.018 | 0.435±0.033 | 0.743±0.027 |
| Dice(12 edges) | 0.593±0.031 | 0.203±0.014 | 0.470±0.008 | 0.547±0.020 | 0.486±0.028 | 0.515±0.025 |
| Dice(30 edges) | 0.490±0.038 | 0.319±0.019 | 0.820±0.014 | 0.860±0.016 | 0.694±0.024 | 0.359±0.028 |
| Pouring(1st 100ml) | 0.441±0.027 | 0.365±0.045 | 0.488±0.019 | 0.624±0.028 | 0.585±0.040 | 0.622±0.034 |
| Pouring(2nd 100ml) | 0.162±0.016 | 0.543±0.044 | 0.818± 0.019 | 0.810±0.018 | 0.491±0.038 | 0.567±0.024 |
| Pouring(3rd 100ml) | 0.232±0.020 | 0.469±0.040 | 0.717±0.017 | 0.438±0.016 | 0.441±0.041 | 0.235±0.025 |
| Shaking(100ml) | 0.630±0.031 | 0.911±0.013 | 0.0576±0.006 | 0.534±0.038 | 0.532±0.039 | 0.592±0.047 |
| Shaking(200ml) | 0.706±0.031 | 0.260±0.013 | 0.236±0.011 | 0.642±0.011 | 0.678±0.032 | 0.345±0.032 |
| Shaking(300ml) | 0.295±0.034 | 0.112±0.009 | 0.606±0.032 | 0.842±0.012 | 0.478±0.027 | 0.568±0.04 |

Tab. 11: **Quantitative results of the mean value and standard error of the mean (SEM) of the twelve acoustic vibration signal descriptors in characterization experiments.**

