# OpenReview forum: "SonicSense: Object Perception from In-Hand Acoustic Vibration"
_robot-learning.org/CoRL/2024/Conference — CoRL 2024_

### Official Review · Reviewer_UVNB · 2024-07-18
**SonicSense: Object Perception from In-Hand Acoustic Vibration**

**Originality:** 3
**Technical Quality:** 3
**Clarity Of Presentation:** 3
**Potential Impact:** 2
**Recommendation:** 2
**Confidence:** 3

**Review:**

The authors introduce a robot hand with embedded contact microphones and use it for object perception: including object identification, 3d shape reconstruction, and container volume differentiation. They collect example data through a exploration policy outlined here, and finally conclude with a learned method to fuse their system data to infer object properties.

The authors aim to expand on previous work by increasing the number of object classes to identify to 83 unique items, covering a range of materials and shapes from real-world and 3d printed objects. They present three different networks for using the audio data for (1) material classification out of 9 classes, (2) 3d pose reconstruction from sparse contact points, and (3) object classification out of 83 (unclear) classes. They also extract 12 manually defined features for ~3 class container volume classification.

The paper is mostly clear and well-written, I can follow the experimental design and results with just a few questions. The video supplement is a useful addition, covering related works and limitations. Overall, this work appears to be several classifications tasks given audio input, which is technically sound but not a huge step beyond previous work. They claim advancement through the combination of multiple perception capabilities, but as each appears to be an individual network, I am not convinced of the contribution.

**Quality Of The Limitations Section:**

3

**Questions For Rebuttal:**

The authors fix the objects to the table surface before collecting data samples. Does this affect the 'realism' of the audio signal collected for more real-world scenarios? To confirm, I understand and support this choice for data collection, but I am curious to characterize what artifacts this may add to your samples beyond discussion from limitation. Have the authors been able to compare the fixed vs not-fixed audio samples?

The authors are particular about the object train/test splits not containing any overlap - I agree. To support that claim, the authors should outline their defined objects train/test/validation splits across the objects in the Appendix at the very least. I think it will help readers understand how much variety there was between the splits.

The robustness to ambient noise section and Figure 6 do not need to be included in the main text, it is a well-known property for contact microphones and not a core focus or contribution of the work.

The authors use both mel-spectrograms and sparse contact points to identify the objects. Can you ablate these inputs to understand how much of the information can be accounted per input?

**Robotics Focus:**

4

**Summary Of Paper:**

The authors introduce a robot hand with embedded contact microphones and use it for object perception: including object identification, 3d shape reconstruction, and container volume differentiation. They collect example data through a exploration policy outlined here, and finally conclude with a learned method to fuse their system data to infer object properties.

**Summary Of Recommendation:**

This paper appears to combine various concepts for audio-based classifiers that have been previously demonstrated in other works. The combination of multiple tasks/networks may not be sufficient to warrant interest at the conference.

---

### Official Review · Reviewer_M19m · 2024-07-18
**Interesting paper with a relevant contribution**

**Originality:** 4
**Technical Quality:** 4
**Clarity Of Presentation:** 5
**Potential Impact:** 3
**Recommendation:** 3
**Confidence:** 4

**Review:**

The paper investigates a relevant an active research topic: acoustic sensing for manipulation. It is well written and comprehensible. The video and appendix are useful and support the paper well. It is great that the authors jointly tackle the hardware and software side of the problem and the engineering choices are well motivated and comprehensible. The paper makes a solid contribution.

The only critique I have, is that the paper paints itself as focusing on sound, but the shape reconstruction is not sound based. I don't think this is a major problem, as the paper also argues it presents an integrated hardware and software solution, and it shows what is possible given the presented system. However, it would be nice to see if using sound could improve the shape estimation results. E.g. contact points would likely not be useful to estimate if something is solid or hollow, but I could imagine that this is possible using sound. Also if the authors tried this and shape estimation did not benefit from sound, this would also be an interesting result.

**Quality Of The Limitations Section:**

3

**Questions For Rebuttal:**

- L330: "Unlike previous approaches, our study involves several orders of magnitude more number of real-world objects with complex geometry and heterogeneous materials." -- it is not several orders of magnitude. please downsize this claim...

**Robotics Focus:**

4

**Summary Of Paper:**

This paper presents an integrated hardware and software solution to recognize object material, shape, and instance. The hardware consists of contact microphones attached to four single-jointed fingers. These are used to explore objects with a simple explorative policy. Contact points, computed based on voltage feedback from the actuated joints, are used to estimate object shape. A feature vector based on audio is used to estimate object material. Audio and contact points are jointly used to recognize object identity. The paper reports good results in each task.

**Summary Of Recommendation:**

The paper tackles a relevant problem and makes a solid technical contribution. It could be stronger if they acutally used sound for the shape reconstruction.

---

### Official Review · Reviewer_bAXR · 2024-07-20
**Multimodal sensing using tactile and audio; State Estimation;**

**Originality:** 4
**Technical Quality:** 5
**Clarity Of Presentation:** 5
**Potential Impact:** 3
**Recommendation:** 3
**Confidence:** 4

**Review:**

The setup of a multi-finger robot hand equipped with contact microphones is quite interesting. The dataset, collected using this system from real-world objects, has significant potential for contribution if made open-source. The tasks tested with this dataset, including container inventory status differentiation, material prediction, 3D shape reconstruction, and object re-identification. Additionally, the paper is well-organized and clearly written and easy to read.

Strengths:
- A novel method to tactile perception using in-hand acoustic vibrations
- A diverse dataset of 83 real-world objects are collected and methods are tested on this dataset to demonstrate effectiveness
- The system setup is not expensive and easy to reproduce

Weaknesses:
- Objects need to be fixed on the table, while in many real world scenarios, the interaction are much more complicated.
- Though using a multi-finger robot hand but no complex manipulation skills are involved
- More modalities can be integrated with audio

**Quality Of The Limitations Section:**

3

**Questions For Rebuttal:**

- Another two related works in robot manipulation with audio signals [1, 2]
- As mentioned in last section, the interaction is more complicated in the real world scenarios. This might be a open research question, but could you elaborate more on the current role and future work of audio datasets in robot learning community? Robotic audio datasets (maybe not only audio but among vision, audio, and tactile, audio have the least amounts) nowadays still limited in real world applications. Robot audio data often cannot generalize to more realistic scenarios.

[1] Li, Hao, et al. "See, hear, and feel: Smart sensory fusion for robotic manipulation." arXiv preprint arXiv:2212.03858 (2022).
[2] Du, Maximilian, et al. "Play it by ear: Learning skills amidst occlusion through audio-visual imitation learning." arXiv preprint arXiv:2205.14850 (2022).

**Robotics Focus:**

4

**Summary Of Paper:**

The paper presents SonicSense, an integrated hardware and software system designed to enhance robot object perception through in-hand acoustic vibration sensing. SonicSense employs a heuristic exploration policy to interact with objects and uses end-to-end learning algorithms to fuse vibration signals for inferring object properties. The paper demonstrates the effectiveness of SonicSense using a diverse set of 83 real-world objects with significant advancements over existing acoustic sensing methods.

**Summary Of Recommendation:**

The setup and dataset of this paper could make a potential contribution to the community, therefore I recommend accept this paper.

---

### Decision · Program_Chairs · 2024-09-04

**Decision:**

Accept

**Comment:**

The authors present a good hardware platform for using audio feedback to perceive objects. The task and the setting are interesting. The reviewers have questions about the experiment setting, like how the fixture will influence the audio response-- in other words, more discussion of the generalizability of the method is needed. Also, the authors are encouraged to demonstrate the contribution of the method further compared to existing audio-based object perception.

----

The authors addressed most of the reviewers' questions well. While the reviewers still think the use of contact microphone in object detection is not a new concept, they believe the new dataset and results are beneficial to the community.